# Genetic Identification and Technological Potential of Indigenous Lactic Acid Bacteria Isolated from *Alheira*, a Traditional Portuguese Sausage

**DOI:** 10.3390/foods13040598

**Published:** 2024-02-16

**Authors:** Nathália Fernandes, Ana Sofia Faria, Laís Carvalho, Altino Choupina, Carina Rodrigues, Ursula Gonzales-Barron, Vasco Cadavez

**Affiliations:** 1Centro de Investigação de Montanha (CIMO), Instituto Politécnico de Bragança, Campus de Santa Apolónia, 5300-253 Bragança, Portugal; nathalia@ipb.pt (N.F.); anafaria@ipb.pt (A.S.F.); laismagalhaescarvalho@hotmail.com (L.C.); albracho@ipb.pt (A.C.); rodrigues.carina7@gmail.com (C.R.); ubarron@ipb.pt (U.G.-B.); 2Laboratório para a Sustentabilidade e Tecnologia em Regiões de Montanha, Instituto Politécnico de Bragança, Campus de Santa Apolónia, 5300-253 Bragança, Portugal

**Keywords:** fermented meat, food quality, phylogenetic analysis, traditional food

## Abstract

*Alheira* is a naturally fermented meat sausage traditionally made in the Portuguese region of *Trás-os-Montes*. Lactic acid bacteria (LAB) are the dominant microorganisms in *alheira* and can endow it with various technological properties. This study aimed (1) to characterize technological features and in vitro antimicrobial activity of LAB isolated from *alheira*, and (2) to reveal associations between such phenotypic characteristics and the isolates species identified through amplification and sequencing of the 16S ribosomal gene. Sixty-two LAB isolates were identified and *Enterococcus (E.) faecium* corresponded to 32.3% of isolates, followed by *Leuconostoc (L.) mesenteroides* (19.4%) and *Latilactobacillus (Lb.) sakei* (17.7%), aligning with previous research on traditional Portuguese fermented meat sausages. The phenotypic analysis of LAB isolates indicated diverse acidification capacities, proteolytic activities, and inhibitory effects against foodborne pathogens *Listeria (L.) monocytogenes*, *Salmonella (S.)* Typhimurium and *Staphylococcus (S.) aureus*. Overall, lactobacilli displayed high inhibition activity against the pathogens *S. aureus*, *L. monocytogenes*, and *S.* Typhimurium. Although the mechanisms for the inhibition of pathogen growth need to be further elucidated, these findings enhance our understanding of LAB diversity and functionality in *alheira* sausages, contributing to product safety and quality.

## 1. Introduction

Mediterranean artisanal foods play a significant role in the development of rural regions, allowing and stimulating local commercialization, employment of the rural population, and preservation of local heritage. *Alheira* is a naturally fermented meat sausage traditionally made in the Portuguese region of *Trás-os-Montes*. The production of *alheira* uses various meats (most commonly pork and poultry meat) that are comminuted to form a heterogeneous batter that is then stuffed in pig- or cattle-intestinal casings or in cellulose-based casings [1,2,3].

In addition, the *alheira* sausage undergoes an intermittent cold smoking process lasting 2–8 days that reduces water activity (a_w_) and pH. The sausage then goes through a drying process (1–2 days) carried out at 55–65% relative humidity. The final product composition is usually 35% water, 25% protein, and 45% lipids, with an a_w_ in a range of 0.9–1.0 and an approximate pH of 5.1–5.8 [1,2,3,4].

These conditions characterize a microbiologically unstable product that allows the growth of pathogenic organisms such as *Staphylococcus aureus*, *Salmonella enterica*, and *Listeria monocytogenes*. Fermentation occurs by the action of acid-producing bacteria that lower the pH of the product. These bacteria may occur naturally or can be added as starter cultures [3]. The *alheira* manufacturing process and final composition can vary considerably between regional producers and even within different batches of a factory. These variations stem from differences in the initial formulation, particularly in the quantity and type of meat (such as various carcass parts and different animal meats) as well as variations in fat and bread content. Other quality factors associated with differences in *alheira* composition are the duration of fermentation and the extent of drying [5].

Lactic acid bacteria (LAB) can be described as gram-positive, catalase-negative, and facultative anaerobic that differ in morphology (e.g., cocci, coccobacilli, or rods) and ferment glucose to lactic acid—through either homofermentative or heterofermentative pathways [6]. The conventional method for identifying LAB species involves utilizing phenotypic attributes, such as determining morphology, fermentation patterns, and lactic acid isomer profiles, or genetic methods, such as 16S rRNA sequencing—a gene that contains both well-conserved and less-conserved regions used to sort different genera and species, or species-specific genes to discriminate those that are closely related [7,8].

LAB associated with food are generally restricted to the genera *Carnobacterium*, *Enterococcus*, *Lactobacillus*, *Lactococcus*, *Leuconostoc*, *Oenococcus*, *Pediococcus*, *Streptococcus*, *Tetragenococcus*, *Vagococcus*, and *Weissella* [6]. LAB have been found to be the dominant microorganisms in alheira [5,7]. Importantly, LAB strains, utilized as starter cultures or those developing spontaneously during meat fermentation, must display specific and discernible metabolic attributes. These encompass the capacity to generate acids and aromatic compounds, curb the propagation of harmful microorganisms, enzymatically degrade proteins, and lack virulence factors and toxins [6]. These attributes stand as imperative safeguards, ensuring the elevated quality and safety of the final product.

The objectives of this study were: (1) to conduct a phenotypic characterization of LAB isolated from *alheira*, encompassing technological features such as *in vitro* proteolytic activity, production of L-lactic acid, acidifying potential and antimicrobial activity against *Staphylococcus (S.) aureus*, *Salmonella (S.)* Typhimurium and *Listeria (L.) monocytogenes*; and (2) to identify genotypically the LAB species and reveal any associations with desired phenotypic characteristics.

## 2. Materials and Methods

### 2.1. Sampling

Fifty-eight *alheira* sausages were obtained from thirteen artisanal establishments located in the towns of *Bragança* (BST, BF), *Mirandela* (CM, FN), *Mogadouro* (BPM), *Vimioso* (BV), *Chaves* (C), *Povoa de Lila* (MP), *Vinhais* (CSM, AG, AMO), *Valpaços* (SM) and *Alfaião* (QP), which belong to the Portuguese northeastern region of *Trás-Os-Montes* (Figure 1). The sausages were acquired unpacked at traditional markets and underwent physicochemical and microbiological analysis. It is worth mentioning that *alheira* sausages were obtained from artisanal producers that do not employ starter cultures in their elaboration.

### 2.2. Physicochemical and Microbiological Analysis of Sausages

The casings enveloping the sausages were removed, and the contents were then cut into small pieces and homogenized. Physicochemical analysis of the sausage included measurement of pH, with a probe (Hanna Instruments, HI5522, Woonsocket, RL, USA) and a_w_ with an Aqualab meter (4TE Decagon, Pullman, WA, USA), measured in triplicate. Additional determinations of moisture, dry matter, and ashes were made according to ISO 1442:1997 (direct drying method) [9], ISO 937:2023 [10], and ISO 936:1998 [11], respectively, and the results were expressed in a dry matter basis (%). Finally, measurements of carbohydrates (CHO), fat, and protein contents (%) were carried out in duplicate and expressed in a dry matter basis (%).

The samples (25 g) were homogenized in 225 mL buffered peptone water and ten-fold dilutions were prepared thereof. The aerobic mesophilic count (AMC) was determined in plate count agar (Liofilchem, Teramo, Italy) at 35 °C ± 1 °C for 48 ± 2 h. Previous reports on AMC demonstrate its positive correlation with LAB counts [5,12,13,14].

*S. aureus* was counted on Baird-Parker agar supplemented with egg yolk tellurite incubated at 35 °C for 48 h, following ISO 6888-1:2021 [15] and *Clostridium (C.) perfringens* was determined using tryptone sulfite cycloserine agar (TSCA) with egg yolk supplement and incubated at 35 °C for 24 h, in accordance with the Compendium of Methods for the Microbiological Examination of Foods [16].

Additionally, the *S. enterica* search was performed using tetrathionate broth (610183, Liofilchem, Teramo, Italy), Rappaport-Vassiliadis broth (610175, Liofilchem, Teramo, Italy), Hektoen enteric agar (610021, Liofilchem, Teramo, Italy), bismuth sulfite agar (610301, Liofilchem, Teramo, Italy) and Xylose-Lysine-Deoxycholate (DSHB3011, Alliance Bio Expertise, Bruz, France), according to the Bacteriological Analytical Manual (BAM) for *Salmonella* detection in food matrices from the Food and Drug Administration (FDA) [17], following the additional step of serological confirmation described in the BAM—with the *Salmonella* Latex Kit (Liofilchem, Teramo, Italy). Samples were assigned 1 if positive and 0 if negative, for the statistical analysis. Determinations were carried out in duplicate.

### 2.3. LAB Isolation

In order to isolate LAB from the *alheira* sausages, one-mL volumes from the 10-fold dilutions were incorporated in De Man, Rogosa, and Sharpe (MRS) and M17 selective agars, overlayed with agar 1.2% and incubated at 30 °C for 48 h [18]. Five typical colonies on MRS and M17 agar (each) were selected for purification and incubated at 30 °C for another 48 h in the respective media—a sampling plan that did not seek the major fermentation driver, but to evaluate diversity within samples. Finally, isolates were confirmed by catalase (3% hydrogen peroxide) and gram tests as well as morphologic observation by microscopy. The confirmed isolated LAB were cryopreserved in 25% glycerol at -80 °C until further testing.

### 2.4. Phenotypic Characterization

#### 2.4.1. Antimicrobial Activity

The antimicrobial activity of LAB isolates was tested against three foodborne pathogens: *S. aureus subsp. aureus* ATCC 6538, *S. enterica subsp. enterica* serovar Typhimurium ATCC 43971, and *L. monocytogenes* WDCM 00019 using the agar spot method. Each LAB strain reactivated in MRS or M17 broth after an overnight culture was spotted (3 or 5 µL) on solidified MRS or M17 agar plates, respectively. Then, the plates were covered with 10 mL of BHI broth with 0.75% (*w*/*v*) bacteriological agar seeded with 1 mL of each bacterial indicator strain (separately) at approximately 8 log CFU/mL—pathogenic strains were revived in 10 mL Brain Heart Infusion (BHI) broth for 16 h at 37 °C. The bacterial strains of *L. monocytogenes* went through an additional step of activation in 5 mL of BHI for 16 h at 37 °C. Cultures were then successively inoculated until reaching the concentration mentioned above. After solidification, plates were incubated at temperatures of 37 °C for 16 h or 10 °C for 10 days [12]. The inhibition diameter was measured (mm) in duplicate and a control without the addition of the LAB was used to validate the results. Sixty-two presumptive LAB with the highest antimicrobial capacity at both temperatures were selected. On this subset of strains, the following phenotypic assays were carried out: proteolytic activity, acidifying capacity, and L-lactic acid production.

#### 2.4.2. Proteolytic Activity

For the determination of exocellular proteolytic activity, overnight cultures were spotted (3 µL) on the surface of milk agar (composed of 10% (*w*/*v*) skim milk powder and 2.5% (*w*/*v*) agar) and incubated at 35 °C for 4 days. Proteolytic activity was measured as the diameter of the clear zones around each LAB colony [19].

#### 2.4.3. Acidifying Capacity

To quantify the acidifying capacity, each isolate was reactivated separately in MRS or M17 broth overnight (30 °C, 24 h). Then, a loop of culture was placed in 10 mL of sterile reconstituted skim milk supplemented with yeast extract 0.3% (*w*/*v*) and glucose 0.2% (*w*/*v*) for two successive subcultures (30 °C for 24 h). Sterile reconstituted skim milk (100 mL, initial pH 6.7) was then inoculated with 1 mL of the 24 h activated culture. For the acidification profiling, pH changes were determined using a pH meter (Hanna Instruments, model HI5522, Rhode Island, USA) equipped with a HI1131 glass penetration probe during incubation at 30 °C during 8 h (*t* = 0, 3, 6, 8 h), and after 24 h. For every strain, pH data was fitted to a decay curve to characterize acidification capacity [18]. The following descriptors were extracted from the fitted curves: ΔpH03: pH decrease between *t* = 0 h and *t* = 3 h; ΔpH06: pH decrease between *t* = 0 h and *t* = 6 h; ΔpH36: pH decrease between *t* = 3 h and *t* = 6 h; and pH6: pH at *t* = 6 h.

#### 2.4.4. L-Lactic Acid Production

To quantify the L-lactic acid (g/L) produced by the LAB, isolates underwent a revival process in 10 mL of MRS or M17 broth, followed by incubation at 37 °C for 24 h [20]. Subsequently, the inoculum was transferred to MRS or M17 agar plates, corresponding to the isolation media, with the aim of obtaining pure isolated colonies. These plates were then placed in an anaerobiosis jar and incubated at 30 °C for 48 h. Two isolated colonies were carefully selected and combined with 5 mL of saline solution. The measurement of absorbance at 625 nm was conducted once turbidity reached an estimated 0.5 on the McFarland scale. Samples were adjusted to fall within absorbance values of 0.08–0.13. Following this, 100 µL of the adjusted samples were transferred to 4 mL of MRS broth and incubated at 30 °C for 4 h. After the incubation period, 1 mL of the culture underwent centrifugation for 5 min at 13,000 rpm, and the resulting pellet was discarded. Subsequently, 10 µL of the supernatant was added to 500 µL of deionized water and vortexed. The concentration of L-lactic acid in g/L was determined using the Kit Nzytech L-lactic (NzyTech, Lisbon, Portugal), UV method, following the manufacturer’s instructions. The concentration obtained is the sum of the free and esterified lactic acid, based on the spectrophotometric measurement of NADH formed through the combined action of L-lactate dehydrogenase (L-LDH) and D-alanine aminotransferase [20].

### 2.5. Genotypic Characterization

#### 2.5.1. DNA Extraction

Pure genomic DNA was obtained using the GF-1 Bacterial DNA Extraction Kit (Vivantis, Selangor, Malaysia). LAB isolates were grown in MRS or M17 broth for 24 h at 37 °C, and 3 mL of bacterial culture was centrifuged at 10,000× *g* for 2 min to obtain a pellet [21]. The pellet was resuspended in 80 µL of R1 Buffer and treated with 20 µL of lysozyme (50 mg/mL; Vivantis, Selangor, Malaysia) for 30 min at 37 °C. The cells were centrifuged at 10,000× *g* for 3 min to form the pellet, which was resuspended in 180 µL of R2 Buffer and 3 µL of Proteinase K (10 mg/mL; Vivantis, Selangor, Malaysia) and incubated in a dry bath at 65 °C for 40 min. Then, 3 µL of RNAse A (20 mg/mL; Vivantis, Selangor, Malaysia) was added and incubated at 37 °C for 10 min, and 372 µL of BG Buffer homogeneous solution was added to the sample and incubated at 65 °C for 20 min. The washing of the DNA was conducted with a clean glass filter membrane, to which absolute ethanol (200 µL) and the sample (558 µL) were transferred and centrifuged at 10,000× *g* for 1 min. The membrane was washed with 650 µL of Wash Buffer and centrifuged at 10,000× *g* 1 min two times for removal of residual ethanol. Pure DNA in the membrane was eluted in 30 µL pre-heated TE buffer for 2 min, centrifuged at 10,000× *g* for 2 min, and stored at −20 °C, as recommended by the manufacturer.

#### 2.5.2. 16S rRNA Amplification

The primers used for amplification of the 16S rRNA gene [22,23] were 27f 5′- AGA GTT TGA TCC TGG CTC AG -3′ and 1492r 5′-CTA CGG CTA CCT TGT TAC GA-3′ at 5µM (IDT, Leuven, Belgium), 1X PCR-Buffer (Frilabo, Maia, Portugal), 200 µM of each dNTP in a mix (Frilabo, Maia, Portugal), 1.25 U of DFS-Taq DNA Polymerase (ThermoFisher Scientific, Lisbon, Portugal), and 10 ng/µL of template DNA, adjusted to a 50 µL reaction. The PCR cycle was 94 °C for 2 min, followed by 30 cycles of 94 °C for 10 sec, 62 °C for 20 sec, and 72 °C for 1 min [16]. An 80 mL agarose gel 1% (*w*/*v*) prepared with 1X TAE Buffer and stained with 4.7 µL Ethidium Bromide was used to load the samples—4 µL of PCR product, 1 µL 5X bromophenol blue (Frilabo, Maia, Portugal) and 1 Kb DNA (Frilabo, Maia, Portugal) Ladder (0.1 µg/µL). Electrophoresis was run at 100 V for 45 min, and the fragments (~1.5 Kb) were visualized in ChemiDocTM (BioRad, Amadora, Portugal). DNA bands were cleaned up using the GF-1 PCR Clean-up Kit (Vivantis, Selangor, Malaysia). The volume of samples was adjusted to 100 µL with nuclease-free water and mixed with 500 µL of PCR Buffer. The sample was loaded to a glass filter membrane and centrifuged at 10,000× *g* for 1 min; then the membrane was washed with Wash Buffer (750 µL) and centrifuged at 10,000× *g* for 1 min two times to remove residual ethanol. Pure DNA in the membrane was eluted in 30 µL TE buffer for 2 min, centrifuged at 10,000× *g* for 2 min, and stored at −20 °C. For subsequent sequencing reactions, the purity of the amplicon (2 µL) was measured with the 260/280 nm absorbance ratio (~1.8).

#### 2.5.3. 16S rRNA Sequencing

Sequencing reactions used the BigDyeTM Terminator v3.1 ready reaction mix (ThermoFisher Scientific, Portugal) with 27f and 1492r primers at (3.2 µM), 5X Sequencing Buffer, and nuclease-free water to a final volume of 7 µL and mixed with 3 µL of purified amplicon [22]. Samples were assessed in duplicate reactions. The parameters for the sequencing reaction were 96 °C for 1 min and 25 cycles of 96 °C for 10 s (denature), 62 °C for 5 s (anneal), and 60 °C (extend) for 4 min. For removal of interferences with base calling, samples were purified with 60 µL of SAM/BigDyeXTerminatorTM bead solution (ThermoFisher Scientific, Lisbon, Portugal) and vortexed at 1800 rpm for 20 min. Capillary electrophoresis carried out in the SeqStudio Genetic Analyzer (Applied Biosystems, Porto, Portugal) was run at 12,000 V for 25 s and the final results were analyzed using the Sequencing Analysis Software v7.0 (Applied Biosystems, Porto, Portugal).

### 2.6. Data Analysis

#### 2.6.1. Species Identification

Sequence results were aligned with reference sequences from the National Center for Biotechnology Information (NCBI, Bethesda, MA, USA) using the rRNA/ITS—16S ribosomal RNA sequences database run with the Basic Local Alignment Search Tool (BLAST) algorithm optimized for highly similar sequences [24]. Finally, sequences with identity equal to or higher than 97% were accepted as the best match for the LAB isolate at the species level [25].

#### 2.6.2. Phylogenetic Tree

The phylogenetic tree was plotted using the R software (version 4.3.0, R Foundation for Statistical Computing, Vienna, Austria) [26]. A multiple sequence alignment was performed using the msa package [27] and the ClustalOmega algorithm. A distance matrix was obtained with the seqinr package [28]. The package ape was used to obtain a phylogenetic tree by the Neighbor Joining (NJ) method [29]. Visuals and annotations were made using the ggtree package [30] and the iTools software (version 6.8.0) [31].

#### 2.6.3. Phenotypic Characterization of LAB

Data were divided into three subsets, by foodborne pathogen species: *L. monocytogenes*, *S.* Typhimurium, and *S. aureus*. Principal component analysis (PCA) of each subset was performed to assess the contribution of the antimicrobial, proteolytic, and acidifying capacities to the differentiation of isolates. The function principal from the psych package was used in R software (version 4.3.0, R Foundation for Statistical Computing, Vienna, Austria) [32], where a varimax-rotated solution for two principal components was obtained. Projections of the sample scores onto the span of the principal components were produced by using the function prcomp from the factoextra package. Heatmaps were created using the pheatmap function [33] to find relationship patterns within the samples. The legend of the variables in the PCA is as follows: Mean inhibition diameter (ID) in mm at 10 °C for *L. monocytogenes* (ID_10 °C Listeria), *S.* Typhimurium (ID_10 °C Salmo), and *S. aureus* (ID_10 °C Staphy) and at 37 °C (ID_37 °C Listeria), (ID_37 °C Staphy), and (ID_37 °C Salmo). pH values: pH6 (after 6 h), Δ03 (drop between 0 h and 3 h), Δ06 (between 0 h and 6 h), Δ36 (between 3 h and 6 h). Proteolytic activity (ProteolyticAct) in mm and Concentration of L-lactic acid (LAC) in g/L.

## 3. Results and Discussion

### 3.1. Genetic Identification of Lactic Acid Bacteria Isolates

The results of sixty-two isolates of LAB identified by 16S gene sequencing are shown in Table 1. Among the identified species, *Enterococcus (E.) faecium* was the most common, accounting for 32.3% of the total, followed by *Leuconostoc (L.) mesenteroides* (19.4%) and *Latilactobacillus (Lb.) sakei* (17.7%). Moreover, *Lactiplantibacillus (Lb.) plantarum (6.5%)*, *Pediococcus (P.) pentosaceus (4.8%)*, and *Weissella (W.) viridescens* (1.6%) were found. The composition of LAB in dry fermented sausages predominantly featured *Lactobacillus*, *Enterococcus*, *Pediococcus*, and *Leuconostoc*, as indicated by previous research [34].

Enterococci, especially *E. faecalis* (*n* = 73) and *E. faecium* (*n* = 60), have been previously isolated from dry smoked fermented sausages–*Catalão*, *Chouriço-preto*, *Linguiça*, *Salsichão* and *Paio* [35]. The enzymatic profile of *Enterococcus spp.* was associated mainly with lipid and protein metabolism. For instance, phosphatase acid, naphthol AS-BI-phosphohydrolase, and cystine aminopeptidase were detected in 100% of isolates. Interestingly, none of the isolates produced α-Galactosidase, α-Glucosidase, α-Mannosidase, and α-Fucosidase-enzymes involved in sugar hydrolysis [35]. Reports on the microbiological composition of *alheira* have demonstrated that the genetic classification of LAB isolated from this fermented sausage is heterogeneous. According to a previous study on the evaluation of *alheira* sausages collected from different production plants, the most common species were *Lactobacillus (L.) plantarum* (72 out of 90) and *E. faecalis* (87 out of 159) [36].

In addition, our findings reveal *L. mesenteroides* as one of the most common LAB (12 out of 62). Previous reports on the *alheira* sausage composition have indicated that the counts of LAB identified as belonging to the *L. mesenteroides* species corresponded to only 4 out of 283 isolates [36]. Such variations can be attributed to the sausage’s production—recipe and ingredients, fermentation—smoking/drying, and storage methods. Interestingly, the *L. mesenteroides* species have been demonstrated to display technological attributes, for example, a strain isolated from a traditional Serbian sausage made with garlic, sweet pepper, fat, and pork exhibited the capability to produce bacteriocins with antilisterial activity, suggesting a potential role of this species in enhancing the safety and quality of cured meat products [8].

Moreover, the LAB classified as *Lb. sakei* had a relative abundance of 18% within *alheira* isolates. This specific species was found to be the dominant (40% relative abundance) and the major fermentation driver in a metagenomics assessment of the *Salame Piemonte* meat sausage, an Italian sausage with a short maturation period [37]. However, members of the *Lb. sakei* species are usually found in low numbers in the *alheira* composition [36]. Such a difference could be attributed to the meat used for the preparation of the sausage: while *alheira* meat is cooked, the meat used for *Salame Piemonte* manufacturing is raw.

A phylogenetic tree (Figure 2) was built based on the 16S gene sequences to assess the evolutionary relationships between the LAB isolates. From the results, it was observed that *L. mesenteroides* and *P. pentosaceus* isolated from different geographical locations clustered differently. This data indicates there are strains within the same species associated with different producers. Small-scale meat processing typically involves using meat that is produced and transformed on-site, leading to greater variability among producers. By definition, artisanal food manufacturing and processing have a high hands-on workload, this manual aspect of food manufacturing leads to a less standardized final product.

Furthermore, by analyzing the tree topology, isolates from the same producer and identified as belonging to the same species were distant from one another, suggesting a heterogeneous collection of strains. A case in point is the phylogenetic relationships among *E. faecium* isolates from the CM producer collection. Such a difference implies a potential variation in the manufacturing conditions within batches, denoting that conditions within *alheira* sausages from a particular producer do not lead to a uniform collection of strains [38].

This phenomenon was observed previously in a study where the heterogeneous metagenomic content between three batches of spontaneously fermented dry sausages that were submitted to standard production parameters—recipe, ingredients, and ripening—yielded distinct microbial profiles. The source of the differences was derived from the natural microbial composition of the meat employed in the sausage-making [29]. Since there are no standard manufacturing processes in the production of artisanal *alheira* sausages, small differences are likely to appear between batches from the same producer. Further evaluation of the genomic content from individual strains might reveal specific niche adaptations.

### 3.2. Alheira Physicochemical and Microbiological Analysis

Previous research on the physicochemical properties of *alheira* emphasizes the significant variability in chemical, physical, and sensory attributes from industrial and small-scale producers across localities [39,40,41]. Correlations of the physicochemical and microbiological analysis of a total of 58 *alheira* sausages are compiled in Figure 3a, and a subset of 22 selected samples from which LAB with high antimicrobial activity were isolated (Section 2.4) are shown in Figure 3b. The heatmaps offer a comprehensive visualization of the interrelationships among the assayed attributes. Regarding the protein content as an example, the producers with a lower amount of meat in the recipe used a high amount of fat, a typical case is the sausages manufactured by the AG producer that had a mean protein content of 13.15% and a mean fat content of 48.86%, whilst BV producer sausages were composed of 28.64% of meat and 34.75% of fat. Previous research on the nutritional value of *alheira* sausages has reported an average protein and lipid content of 15% and 35%, respectively [1,2,3,4,5,14].

Moreover, the composition of the fat in the *alheira* was previously classified as monounsaturated fatty acids (MUFAs) and saturated fatty acids (SFAs), although polyunsaturated fatty acids (PUFAs) could be detected in lower amounts [4]. Overall, the fat composition reflects the animal’s diet which is more likely to include fatty components, this is particularly common in swine [41]. The carbohydrate composition was predominantly high across the different producers, which is a consequence of the use of moistened bread in the sausage preparation, these proportions vary according to the amount of meat and fat in the producer’s recipe [2]. The mean carbohydrate content of *alheira* in this study was 39.40% (n = 58), when compared to previous reports this value was considerably higher: 19.46% (n = 40) [2], 21.5% [4], and 15.2% [38].

The *alheira* is considered a high-moisture food product. The mean value of a_w_ in the current study was 0.986 ± 0.008 (n = 58), which causes great concern regarding safety and quality since a high a_w_ value makes the environment favorable for the growth of foodborne pathogens and spoilage microorganisms with minimum a_w_ up to 0.98, which is the case for *Salmonella* spp. 0.94, *Staphylococcus* spp. 0.86, *Clostridium botulinum* 0.94, *L. monocytogenes* 0.97 [42]. The mean pH of the sausages was 4.25 ± 0.23, the highest pH was 4.68 and the lowest was 3.88; previous reports have demonstrated similar values of pH (4.92) [2]. The low pH is a consequence of the fermentation promoted by the LAB naturally occurring in the sausage [6]. Moreover, a low pH value (5.5 or less) results in a decrease in water binding capacity and provides conditions that are less favorable to spoilage [41].

The ash content has been previously shown to be highly correlated (r = 0.88, *p* < 0.001) with NaCl that is used during the seasoning of the meat. In our results, the mean ash content was 3.59 ± 0.87%, comparable to another reference study that found 2.21 ± 1.20% [4]. Additional hurdles used in the making of *alheira* are the exposure to the products of wood combustion that provide protection against spoilage microorganisms. Such by-products confer a bacteriostatic activity mainly driven by formaldehyde—which is the main component of the smoke, aliphatic acids, other aldehydes, and phenolic compounds [41]. These antioxidant compounds contribute to the preservation of the meat by protecting lipids from oxidation.

The microbiological quality of these sausages was determined, and the mean AMC results were 8.80 ± 1.58 log CFU/g. Similar results were obtained from previous research, such as 8.42 ± 1.26 [5], 8.5 ± 0.6 [12], and 8.28 ± 0.67 [13]. AMC is a generic test for mesophiles—that grow at temperatures in the range of 25 to 40 °C. A research study [38] compared the *alheira* microbiological analysis results with the guidelines of microbiological quality for ready-to-eat foods [43]. However, this guideline states microbiological quality limits of AMC from smoked sausages were classified as not appropriate for the measurement of poor hygienic quality. In fact, high microbial counts were previously thought to be involved with *alheira* spoilage; however, a high AMC in *alheira* did not reflect organoleptic signs of deterioration [13]. Taking into consideration the naturally fermented nature of *alheira*, a high AMC is expected. Moreover, previous reports have demonstrated the positive correlation of AMC with LAB counts—the dominant microflora [5,13]. In this study, the microbiological counts of *Lactobacillus spp.* were 9.83 ± 1.03 log CFU/g and *Lactococcus spp.* were 9.22 ± 1.32 log CFU/g. AMC has been shown to double in *alheira* from the immediate preparation and filling manufacture step to the final smoked dry sausage [12].

Regarding microbiological safety, the mean counts of *Staphylococcus spp.* were 2.46 ± 1.40 log CFU/g, with a minimum value of 0.699 log CFU/g and a maximum value of 6.06 log CFU/g. Eleven out of fifty-eight sausages were considered unacceptable and potentially hazardous since *Staphylococcus* spp. counts were superior to 4 log CFU/g [43]. Similar results have been previously reported in *alheira* analysis, mainly attributed to the manual handling of the sausage components during filling. The pathogen occurrence could also arise from cross-contamination from initially contaminated meat carcasses, equipment, or utensils [12,13,14].

The mean *Clostridium* spp. counts were 1.07 ± 0.94 log CFU/g with minimal values of 0.699 log CFU/g and the maximum was 4.127 log CFU/g. Eight out of fifty-eight sausages were unsatisfactory in terms of microbiological safety once the colony counts had values higher than 2 log CFU/g. Moreover, one sausage was considered potentially hazardous with a 4.13 log CFU/g pathogen load. These results are to be viewed with caution in the context of public health [43]. Previous reports on *alheira* microbiological safety have shown mean values of 1.2 ± 0.5 log CFU/g, which were in the acceptable range [5]. Pathogenic microorganisms including *C. perfringens*, *Salmonella* spp., and *S. aureus* are the most prevalent in *alheira* sausages [13]. The sporulated form of *C. perfringens* is resistant to extreme temperatures, similar to the enterotoxin produced by *Staphylococcus aureus.* The high levels of contamination observed after analysis of samples are related to poor hygienic and sanitary manufacturing conditions.

### 3.3. Lactic Acid Bacteria Phenotypic and Genetic Analysis

The phenotypic and genetic features of sixty-two LAB isolates are shown in Figure 4, and the summary statistics are in Appendix A. The phenotypic analysis of LAB indicated diverse acidification capacities. For instance, *E. faecium* (n = 20) had the steepest (0.516) pH drop between time intervals 3 and 6 h, while *Lb. herbarum* had a pH drop of 0.056 during the same period. Previous research on the acidifying ability of enterococci has demonstrated that the more rapid acidifiers were from strains of food origin [44]. However, the collection of strains selected in this study appears to behave as slow fermenters since the reduction of pH was achieved by only 0.3–0.9 units.

In the case of *alheira*, the volatile acids derived from the smoke have an additional role in reducing the pH, which influences the metabolic traits of fermenter LAB [41]. The rate of pH fall has an equally if not more important role than the final pH in determining the physical properties of the meat. The glycolysis performed by LAB in the *alheira* positively influences the preservation of the product by lowering the pH. A low pH value and a_w_ at the beginning of the processing minimizes the proliferation of pathogens and detrimental bacteria [1].

In our study, *E. faecium* was the most abundant LAB. However, the presence of enterococci in the *alheira* and other traditional Portuguese fermented sausages has been pointed out as controversial due to enterococci’s behavior as opportunistic pathogens. In fact, *E. faecium* isolated from these sausages was associated with resistance to antibiotics such as erythromycin (36.4%) and tetracycline (2.3%). This resistance could hinder the effective treatment of nosocomial infections associated with enterococci [45].

Regarding the inhibitory effects displayed by the assayed LAB against foodborne pathogens, lactobacilli presented a high (<10 mm) inhibition diameter (ID) against the pathogens *S. aureus*, *L. monocytogenes*, and *S.* Typhimurium. The isolates identified as *Lb. plajomi* and *Lb. plantarum* could inhibit *L. monocytogenes* at 10 °C with IDs of 21.62 and 19.77 mm, respectively. However, when tested under a higher temperature (37 °C), the isolates with antilisterial activity were identified as *Lb. herbarum* (ID = 11.79 mm) and *Lb. sakei* (ID = 11.35 mm). These findings indicate a variation in the most effective antilisterial strain according to the temperature, the inhibition could then be associated with the action of enzymatic compounds. In fact, the production of antibacterial compounds is a natural way of preserving meat and can be produced in combination with organic acids [46,47,48]. Moreover, the results of lactic acid production indicate that the inhibitory action of organic acids is likely to have contributed to the antilisterial activity displayed by these lactobacilli strains which yielded 0.4–0.5 g/L of lactic acid.

Similar patterns were observed in LAB antimicrobial activity against *S. aureus*. The anti-staphylococcal activity was mainly promoted by *Lb. plajomi* (ID = 11.91 mm) and *L. herbarum* (ID = 11.29 mm) at 10 °C. While at 37 °C, inhibition of antimicrobial growth was promoted by *Lb. paracasei* (ID = 9.06 mm) and *Lb. plantarum* (ID = 8.29 mm). The antimicrobial activity of *L. mesenteroides* against *S.* Typhimurium remained consistent at both temperatures. At 10 °C, *Lb. herbarum* (ID = 11.21 mm) and *L. mesenteroides* (ID = 10.89 mm) exhibited the highest inhibition. Similarly, at 37 °C, the inhibition was displayed especially by *L. mesenteroides* (ID = 12.89 mm) and by *Lb. plantarum* (ID = 11.56 mm).

The application of LAB as protective cultures is proven effective in prolonging the shelf life of meat and meat products [6,46,47,48]. Previous reports have demonstrated the use of LAB from *alheira* as biocontrol agents. For instance, a partially purified bacteriocin produced by *Lb. plantarum* 9A3 was added to a culture of *L. monocytogenes* during its exponential phase and effectively repressed pathogenic growth by 3.57 log CFU/mL for a 12 h period [48]. Moreover, a *P. acidilactici* HA-6111-2 producer of pediocin PA-1 presented antimicrobial activity against *L. innocua* when added (10^9^ CFU/mL) as an inoculum in an artificially contaminated *alheira* paste with 10^6^ CFU/mL of *L. innocua*. The bacteriocinogenic LAB could inhibit the growth of the pathogen from an initial population of 4.5 log CFU/g to 2 log CFU/g after 54 days at cold storage [47].

Regarding the proteolytic activity displayed by LAB, it has been previously observed that the production of proteases influences changes in the texture and aroma of the meat [49]. This influence can be negative, for instance, the products of protein degradation such as mercaptans, amines, and fatty acids result in foul odors and flavors indicative of the development of putrefaction and rancidity [41]. In the case of *alheira*, the ingredients (meat, spices, and condiments) are boiled in water at 100 °C for 30 min. During this process, many physicochemical and sensorial changes occur in the meat, for instance, the pH increases, the collagen softens, and the typical aroma develops by the release of volatile sulfur-containing compounds, generated by the degradation of amino acids such as cysteine and methionine [14,41]. Other amino acids such as leucine, phenylalanine, and isoleucine have been associated with the acidic flavor [50].

Previous reports on the proteolytic activity displayed by lactobacilli indicate that most species can bring about proteolysis but to different extents [49]. Our findings demonstrate that the highest proteolytic activity (3.46 mm) was promoted by *Lb. paracasei* strains, while *Lb. herbarum* had the lowest value (0.41 mm) among lactobacilli. In a previous study, 133 LAB isolated from a Chinese fermented sausage were screened for proteolytic activity [51]. Most of the strains (63.16%) could effectively hydrolyze myofibrillar proteins and 57.74% sarcoplasmic proteins. These strains were identified as *Lb. plantarum*, *Lb. pentosus* and *Lb. fermentum*. Moreover, proteolytic enzymes have been used to improve meat sensorial attributes, shorten maturation time, and delay lipid oxidation [49].

To undertake a more comprehensive investigation into the variability of physicochemical attributes exhibited by LAB, a principal components analysis (PCA) was employed. Isolates were annotated with their species-level genetic identification. The dataset was segregated to account for the presence of the three foodborne pathogens that were the subject of analysis, yielding three distinct PCA maps (depicted in Table 2).

For each map, two principal components were retained (based on their eigenvalues surpassing 1), accounting for a substantial portion of the variation (approximately 80%). In the context of Figure 5, an analysis was conducted on a subset of LAB with antimicrobial activity against *S.* Typhimurium at 37 °C and 10 °C (PC1: 66.2%; PC2: 16.2%).

The variables ΔpH36, ΔpH06, pH6, and the inhibition diameter at both temperatures presented the highest loadings, this component primarily measured the antimicrobial activity and acidifying capacity of the LAB. Moreover, there was a stronger positive correlation with the variables ΔpH03 (0.212) and proteolytic activity (0.609) within the second component compared to the lactic acid (−0.771) yield (Table 2).

A similar pattern was observed for *S. aureus* (Figure 6; PC1: 62.4%; PC2: 16.1%) and *L. monocytogenes* assessments (Figure 7; PC1: 64.4%; PC2: 15.2%). However, the influence of different temperatures on inhibition diameters was more evident in the latter, as indicated by the second component, at 37 °C (0.387) and 10 °C (−0.099). The overarching acidifying capacity was considerably attributed to the action of *E. faecium*.

## 4. Conclusions

The careful selection of lactic acid bacteria as starter cultures or biological food-grade preservatives in meat sausages is pivotal for regulating the fermentation process—ensuring product safety by inhibiting foodborne pathogens while contributing to the desired organoleptic characteristics, enhancing consistency and quality. The absence of standardized production processes for the *alheira* sausage may lead to a microbiologically unstable product, causing variations in quality. The implementation of management practices for food production with a standardized approach focused on enhancing quality and safety could potentially elevate the market choices for artisanal products. These products, integral to Mediterranean culture, biodiversity, and economy, require thorough investigation and protection. The observed batch-to-batch variations in the microbial profiles indicate that conditions within *alheira* sausages from a specific producer do not lead to a uniform collection of strains. These variations have implications for product consistency and may need further investigation into genetic heterogeneity. A standardized approach could include the use of an ad-hoc starter culture. In this study, sixty-two LAB isolates were identified, with *E. faecium* (32.3%), *L. mesenteroides* (19.4%), and *Lb. sakei* (17.7%) as the most prominent species, consistent with previous research on Portuguese fermented meat sausages. Phenotypic analysis showed diverse acidification capacities, proteolytic activities, and inhibitory effects against *L. monocytogenes*, *S.* Typhimurium, and *S. aureus* mainly displayed by lactobacilli. Upcoming research will work on further characterizing the mechanism of inhibition expressed by these strains (e.g., bacteriocins, organic acids, proteases/peptidases), and explore their use as fermentation drivers. This study gathers important data for understanding the composition of *alheira* and its intrinsic properties, which in the future could be applied in the development of new pure cultures for the preservation of fermented meat sausages.

## Figures and Tables

**Figure 1 foods-13-00598-f001:**
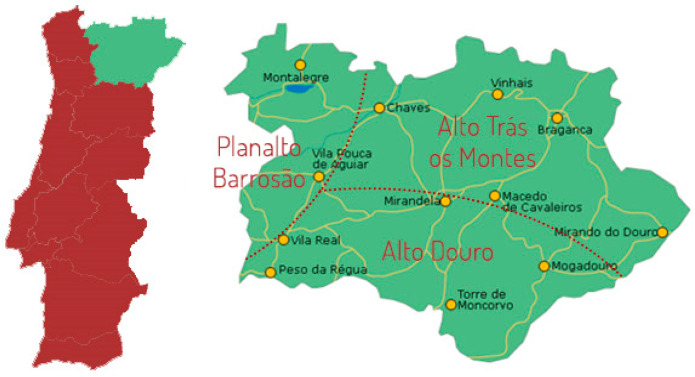
Map of the geographical location of different *alheira* collection sites. The dotted lines indicate three Northern Portuguese regions: Planalto Barrosão, Alto Trás-os-Montes and Alto Douro. Source: *Instituto Nacional de Estatística* (INE).

**Figure 2 foods-13-00598-f002:**
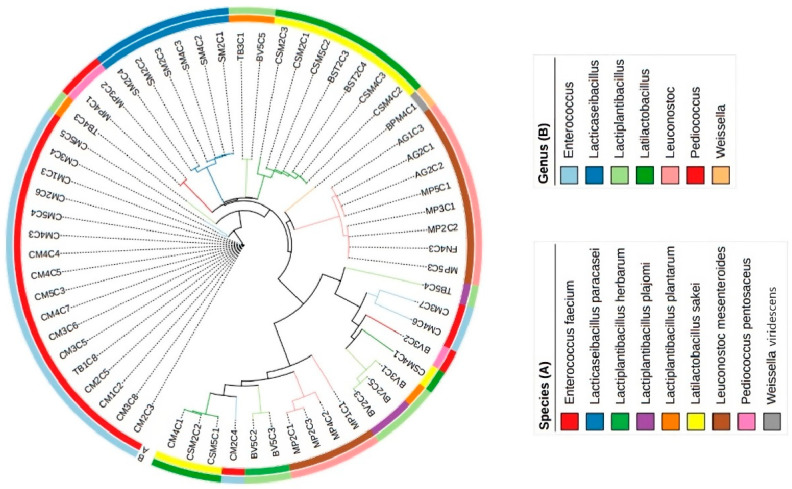
Phylogenetic tree of 16S rRNA sequences from LAB isolates with information at species (**A**) and genus (**B**) level.

**Figure 3 foods-13-00598-f003:**
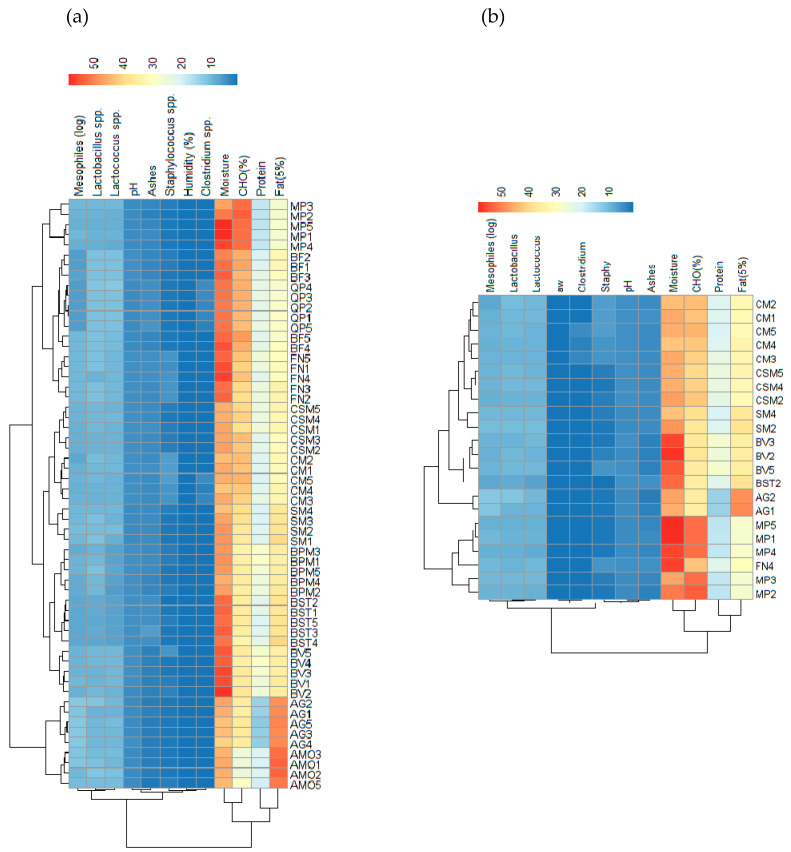
Heatmap of physicochemical and microbiological data of (**a**) total (n = 58) and (**b**) selected (n = 22) *alheira* sausages (n = 59). Legend: The variables are described in Section 2.2.

**Figure 4 foods-13-00598-f004:**
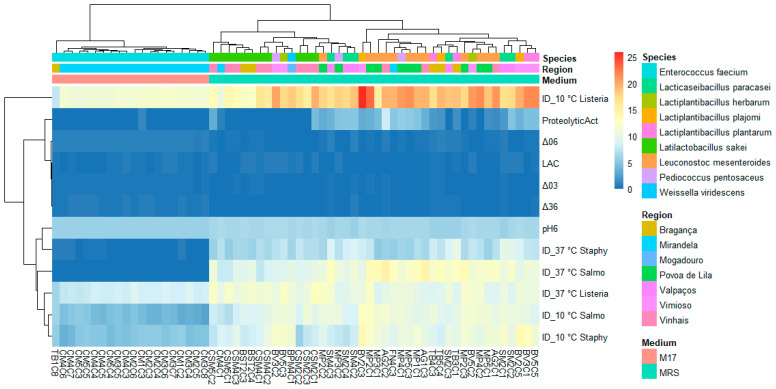
Heatmap of physicochemical characterization data of lactic acid bacteria (LAB) isolated from *alheira* and species identification of LAB isolates. Legend: See Section 2.6.3.

**Figure 5 foods-13-00598-f005:**
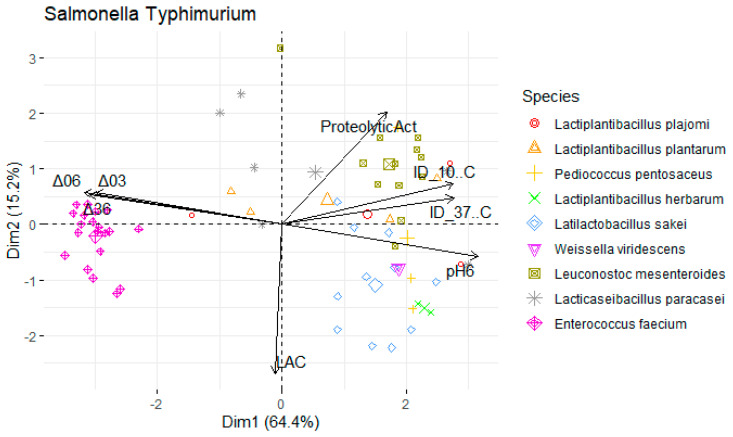
Map of the first and second principal components of the tested technological properties of lactic acid bacteria (LAB) against Salmonella Typhimurium isolated from *alheira* sausage and species identification of LAB isolates. Legend: See Section 2.6.3.

**Figure 6 foods-13-00598-f006:**
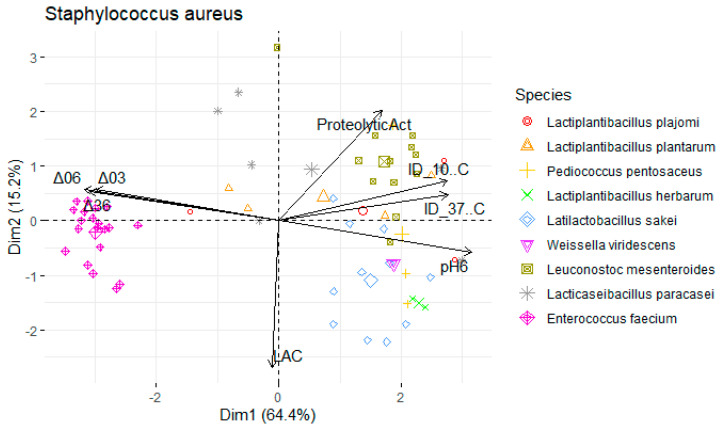
Map of the first and second principal components of the tested technological properties of lactic acid bacteria (LAB) against Staphylococcus aureus isolated from *alheira* sausage and species identification of LAB isolates. Legend: See Section 2.6.3.

**Figure 7 foods-13-00598-f007:**
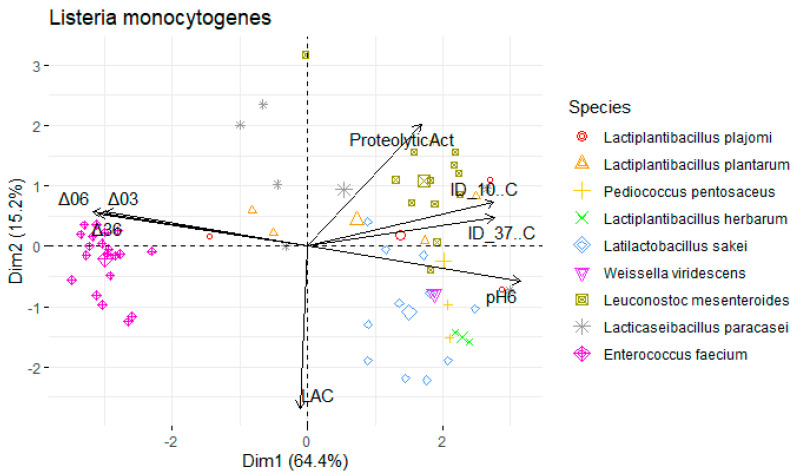
Map of the first and second principal components of the tested technological properties of lactic acid bacteria (LAB) against Listeria monocytogenes isolated from *alheira* sausage and species identification of LAB isolates. Legend: See Section 2.6.3.

**Table 1 foods-13-00598-t001:** Identification results of 62 isolates of LAB by 16S gene sequencing. The main identity expressed is the mean percentage and in brackets are the [minimum; maximum] individual values.

N. Isolates	Species	GenBank_ID	Identity (%)
20	*Enterococcus faecium*	NR_115764.1	99.2 [98.4;100.0]
6	*Lacticaseibacillus paracasei*	NR_025880.1	100.0 [100;100.0]
2	*Lactiplantibacillus herbarum*	NR_145899.1	99.5 [99.3;99.6]
3	*Lactiplantibacillus plajomi*	NR_136785.1	98.9 [98;99.8]
4	*Lactiplantibacillus plantarum*	NR_042394.1	98.5 [97;100.0]
11	*Latilactobacillus sakei*	NR_115172.1	99.8 [99.6;100.0]
4	*Leuconostoc mesenteroides*	NR_074957.1	98.5 [97;100.0]
8	*Leuconostoc mesenteroides subsp. jonggajibkimchii*	NR_157602.1	99.9 [99.8;100.0]
3	*Pediococcus pentosaceus*	NR_042058.1	99.5 [99.4;99.6]
1	*Weissella viridescens*	NR_040813.1	100.0

**Table 2 foods-13-00598-t002:** Map of the first and second principal components of the tested technological properties of LAB isolated from *alheira* sausage and species identification of LAB isolates. Legend: See Section 2.6.3.

Variables	*S.* Typhimurium		*L. monocytogenes*		*S. aureus*	
	PC1	PC2	PC1	PC2	PC1	PC2
ProteolyticAct	0.538	0.609	0.494	0.653	0.508	0.612
pH6	0.942	−0.232	0.960	−0.159	0.950	−0.175
ΔpH03	−0.896	0.212	−0.903	0.137	−0.901	0.173
ΔpH06	−0.956	0.225	−0.968	0.152	−0.961	0.173
ΔpH36	−0.927	0.218	−0.940	0.149	−0.931	0.162
LAC	−0.068	−0.771	−0.032	−0.783	−0.032	−0.812
ID_10C	0.884	0.209	0.758	0.387	0.834	0.218
ID_37C	0.872	0.299	0.784	−0.099	0.834	0.143
% of variance	66.3	16.2	62.4	16.1	64.4	15.2
Cumulative % of var.	66.3	82.447	62.4	78.5	64.4	79.6

## Data Availability

The data are not publicly available due to further work being developed using the same bacterial strains.

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
