# Peer review of "Genetic Identification and Technological Potential of Indigenous Lactic Acid Bacteria Isolated from Alheira, a Traditional Portuguese Sausage"

_foods, 2024, doi:10.3390/foods13040598_

Round 1

Reviewer 1 Report

Comments and Suggestions for Authors

The study entitled "The Genetic Identification and Technological Potential of  Indigenous Lactic Acid Bacteria Isolated from Alheira, which is A Traditional Portuguese Sausage. The study demonstrated that Lactic Acid Bacteria (LAB) are the dominant microorganisms in alheira, and it characterized the technological features and in vitro antimicrobial activity of LAB isolates which were genotypically identified to species level through amplification and sequencing of the 16S ribosomal gene. The study also indicated that the phenotypic analysis of LAB isolates indicated diverse acidification capacities, proteolytic activities, and inhibitory effects against foodborne pathogens.  Lactobacilli presented a higher inhibition diameter against the pathogens Staphylococcus aureus, Listeria monocytogenes, and Salmonella Typhimurium. The study is helpful for understanding of LAB diversity and functionality in alheira sausages, contributing to product safety and quality assessment.

The study is well designed and written well, however English editing for correction of some grammar mistakes is required.

Comments on the Quality of English Language

The study entitled "The Genetic Identification and Technological Potential of  Indigenous Lactic Acid Bacteria Isolated from Alheira, which is A Traditional Portuguese Sausage. The study demonstrated that Lactic Acid Bacteria (LAB) are the dominant microorganisms in alheira, and it characterized the technological features and in vitro antimicrobial activity of LAB isolates which were genotypically identified to species level through amplification and sequencing of the 16S ribosomal gene. The study also indicated that the phenotypic analysis of LAB isolates indicated diverse acidification capacities, proteolytic activities, and inhibitory effects against foodborne pathogens.  Lactobacilli presented a higher inhibition diameter against the pathogens Staphylococcus aureus, Listeria monocytogenes, and Salmonella Typhimurium. The study is helpful for understanding of LAB diversity and functionality in alheira sausages, contributing to product safety and quality assessment.

The study is well designed and written well, however English editing for correction of some grammar mistakes is required.The study entitled "The Genetic Identification and Technological Potential of  Indigenous Lactic Acid Bacteria Isolated from Alheira, which is A Traditional Portuguese Sausage. The study demonstrated that Lactic Acid Bacteria (LAB) are the dominant microorganisms in alheira, and it characterized the technological features and in vitro antimicrobial activity of LAB isolates which were genotypically identified to species level through amplification and sequencing of the 16S ribosomal gene. The study also indicated that the phenotypic analysis of LAB isolates indicated diverse acidification capacities, proteolytic activities, and inhibitory effects against foodborne pathogens.  Lactobacilli presented a higher inhibition diameter against the pathogens Staphylococcus aureus, Listeria monocytogenes, and Salmonella Typhimurium. The study is helpful for understanding of LAB diversity and functionality in alheira sausages, contributing to product safety and quality assessment.

The study is well designed and written well, however English editing for correction of some grammar mistakes is required.The study entitled "The Genetic Identification and Technological Potential of  Indigenous Lactic Acid Bacteria Isolated from Alheira, which is A Traditional Portuguese Sausage. The study demonstrated that Lactic Acid Bacteria (LAB) are the dominant microorganisms in alheira, and it characterized the technological features and in vitro antimicrobial activity of LAB isolates which were genotypically identified to species level through amplification and sequencing of the 16S ribosomal gene. The study also indicated that the phenotypic analysis of LAB isolates indicated diverse acidification capacities, proteolytic activities, and inhibitory effects against foodborne pathogens.  Lactobacilli presented a higher inhibition diameter against the pathogens Staphylococcus aureus, Listeria monocytogenes, and Salmonella Typhimurium. The study is helpful for understanding of LAB diversity and functionality in alheira sausages, contributing to product safety and quality assessment.

The study is well designed and written well, however English editing for correction of some grammar mistakes is required.The study entitled "The Genetic Identification and Technological Potential of  Indigenous Lactic Acid Bacteria Isolated from Alheira, which is A Traditional Portuguese Sausage. The study demonstrated that Lactic Acid Bacteria (LAB) are the dominant microorganisms in alheira, and it characterized the technological features and in vitro antimicrobial activity of LAB isolates which were genotypically identified to species level through amplification and sequencing of the 16S ribosomal gene. The study also indicated that the phenotypic analysis of LAB isolates indicated diverse acidification capacities, proteolytic activities, and inhibitory effects against foodborne pathogens.  Lactobacilli presented a higher inhibition diameter against the pathogens Staphylococcus aureus, Listeria monocytogenes, and Salmonella Typhimurium. The study is helpful for understanding of LAB diversity and functionality in alheira sausages, contributing to product safety and quality assessment.

The study is well designed and written well, however English editing for correction of some grammar mistakes is required.The study entitled "The Genetic Identification and Technological Potential of  Indigenous Lactic Acid Bacteria Isolated from Alheira, which is A Traditional Portuguese Sausage. The study demonstrated that Lactic Acid Bacteria (LAB) are the dominant microorganisms in alheira, and it characterized the technological features and in vitro antimicrobial activity of LAB isolates which were genotypically identified to species level through amplification and sequencing of the 16S ribosomal gene. The study also indicated that the phenotypic analysis of LAB isolates indicated diverse acidification capacities, proteolytic activities, and inhibitory effects against foodborne pathogens.  Lactobacilli presented a higher inhibition diameter against the pathogens Staphylococcus aureus, Listeria monocytogenes, and Salmonella Typhimurium. The study is helpful for understanding of LAB diversity and functionality in alheira sausages, contributing to product safety and quality assessment.

The study is well designed and written well, however English editing for correction of some grammar mistakes is required.The study entitled "The Genetic Identification and Technological Potential of  Indigenous Lactic Acid Bacteria Isolated from Alheira, which is A Traditional Portuguese Sausage. The study demonstrated that Lactic Acid Bacteria (LAB) are the dominant microorganisms in alheira, and it characterized the technological features and in vitro antimicrobial activity of LAB isolates which were genotypically identified to species level through amplification and sequencing of the 16S ribosomal gene. The study also indicated that the phenotypic analysis of LAB isolates indicated diverse acidification capacities, proteolytic activities, and inhibitory effects against foodborne pathogens.  Lactobacilli presented a higher inhibition diameter against the pathogens Staphylococcus aureus, Listeria monocytogenes, and Salmonella Typhimurium. The study is helpful for understanding of LAB diversity and functionality in alheira sausages, contributing to product safety and quality assessment.

The study is well designed and written well, however English editing for correction of some grammar mistakes is required.The study entitled "The Genetic Identification and Technological Potential of  Indigenous Lactic Acid Bacteria Isolated from Alheira, which is A Traditional Portuguese Sausage. The study demonstrated that Lactic Acid Bacteria (LAB) are the dominant microorganisms in alheira, and it characterized the technological features and in vitro antimicrobial activity of LAB isolates which were genotypically identified to species level through amplification and sequencing of the 16S ribosomal gene. The study also indicated that the phenotypic analysis of LAB isolates indicated diverse acidification capacities, proteolytic activities, and inhibitory effects against foodborne pathogens.  Lactobacilli presented a higher inhibition diameter against the pathogens Staphylococcus aureus, Listeria monocytogenes, and Salmonella Typhimurium. The study is helpful for understanding of LAB diversity and functionality in alheira sausages, contributing to product safety and quality assessment.

The study is well designed and written well, however English editing for correction of some grammar mistakes is required.

Reviewer 2 Report

Comments and Suggestions for Authors

  In the submitted manuscript, the authors have tried to reveal associations between LABs isolated from many kinds of Alheria and their phenotypic characteristics.  Although the concept of the submitted manuscript is of interest, however, results and discussions provided in the manuscript do not seem to satisfy the criteria as a scientific journal.  I have concerns to the manuscript with following insufficient points:

  There are possibilities on the inhibition of the growth of the pathogens by presence of organic acids (acetic acid, lactic acids, and other SCFA), proteases/peptidases, and antibacterial compounds. The authors have not confirmed the contribution of those.  Further, the “isolatable” LAB strains do not always reflect the environment of their origin, thus the authors should perform microbiome analysis instead of traditional isolation and identification of the strains.  Surely, the isolation process and isolated strains themselves are also necessary and will provide the authors with important information to reveal the characteristics they want to.

Comments on the Quality of English Language

There are some typographical errors.

Reviewer 3 Report

Comments and Suggestions for Authors

This manuscript describes a substantial body of work attempting to characterize the lactic acid bacteria (LAB) responsible for the fermentation of a regional fermented meat sausage.  The basic approach using both cultural and metagenomic techniques is in keeping with modern approaches to understanding populations interactions associated with microbial communities.  However, there are several technical issues that need to be addressed.  Additionally, there are several editorial issues that need to be addressed.  Examples of both are provided below.

Line 26.  The use of the word “higher” immediately generates the need to answer the question “higher than what.” 

Line 31.  Words that appear in the title should not be repeated as key words.

Lines 35-36.     Why are Alheira and Tras-os-Montes italicized?

Lines 36-37.  Modify “and the final product is as paste” to “…are comminuted to form an emulsion which is then…"

Line 39.  Need to provide more detail.  Is this a "cold smoked" or “hot smoked” process?  Do the LAB remain viable?  Does the LAB increase or decrease during the process?

Line 45.  It would be helpful to provide a simple definition of what a Lactic Acid Bacterium is. 

Lines 46-47.  Delete the phrase “,,,similar in morphology, metabolism and are related phylogenetically…”.  The authors spend the rest of the sentence proving that this part of the sentence is incorrect.

Line 84.  It is unclear what the authors mean by “partitioned.

Line 99.  The authors should replace Salmonella spp. with Salmonella enterica.  There are only two species of Salmonella, S. enterica and S. bongorii.  Most Salmonella encountered in foods are S. enterica.

Lines 103-104. The phrase “with an additional step of serological confirmation” requires one or more citations outlining the methods used.

Lines 110-111.  This protocol indicates that the authors have made an unstated assumption that the LAB responsible for the fermentation of the meat is the most numerous.  There are many examples where the major drivers of good fermentations are not the most numerous.  By only picking 5 colonies, if there is a 10-fold differential between the most numerous and the important fermentation driver, the odds of getting the one with a lower prevalence is small. Furthermore, these fermentations often sequential in nature where the predominant strain changes as the acidity of the food increases.  Classic examples include products such as sauerkraut, pickles, and fermented meats.  Was also quite surprised to see the lack of Pediococcus species which are common in fermented meats.

Line 127.  Were the Salmonella and S. aureus employed actually capable of growing a 10 C at these pH levels and water activities?

Lines 176-177.  I am surprised that the extracted DNA was stored at -20 C instead of -80 C.  Do the authors have any information on the stability of the extracted DNA?

Lines 276-277.  It is unclear what the phrase “6.5, 4.8, and 1.6% of the total population” refers to?  Is it the percentage of time these genera were the predominant isolate from the meat or the overall times of times the genera were isolated (concentration vs. prevalence).

Table 1.  Do not leave the one species blank.

Line 337.  It is unclear whether the isolates from the current study where actually evaluated for bacteriocin production or the presence of the genes for the synthesis.

Lines 383-384.    The antimicrobial activity of organic acids is dependent on the molecule being in its undissociated form.  Using the pH of the environment and the pKa of lactic acid, need to calculate the amount of undissociated lactic acid.  Also, it would helpful to also know what other organic acids are produced by the different isolates.

Conclusions.  Almost all of this section is a restatement of the findings or discussion and not true conclusions.  Considering the state of the research, the authors should consider deleting this section.

References.  The authors should review the journals “style guide” and then ensure that they consistently follow it.  In particular, the capitalization of article titles is very inconsistent.

Comments on the Quality of English Language

It is generally sound but there are opportunties to eliminate material that not directly needed to produce a "tight" manuscript.

Reviewer 4 Report

Comments and Suggestions for Authors

The manuscript is characterized by interesting, extensive, and complex contents. It, however, needs some changes, based on the following observations.

ABSTRACT: It would be better to list bacterial species after the phrase ".... against food-borne pathogens" (e.g. in brackets)

References numbering in the manuscript: see the "Instructions for Authors".

LAB: this acronym should be checked throughout the manuscript (i.e. LAB  or LABs)

Line 38: Ferreira et al, 2006 [1] report “cattle intestinal casings” and not “pig intestinal casings”. It would be better to find and write a different reference. Esteves et al.,2008 [1] refer to “pig casings”

Line 39: Ferreira et al, 2006 [1] refer to “smoking for 2-8 days”

Lines 41-42: [1–3] must be [1–2]. It is recommended to follow the "Instructions for Authors". Reference [2] is not reported in the text.

Line 49: see the comments for lines 41-42.

Line 92: the amount of buffered peptone water should be reported in the text.

Line 93: “Total mesophiles were determined in Plate Count Agar at 35 °C for 48 h…”. It would be more appropriate to write “Aerobic Mesophilic Count, Total Plate Count, or Aerobic Colony Count”. Plates are incubated at 35°C ± 1°C for 48 ± 2 h

The results of this test mainly indicate the hygienic quality of raw material, storage conditions, and the product shelf life.

I ask the authors why this test was carried out. Alheira is a naturally fermented meat sausage, so the aerobic mesophilic count (AMC) could not be useful in checking the hygienic quality of the product. In addition, Lactic Acid Bacteria (LABs), as the dominant microorganisms in Alheira sausage, can interfere with the AMC in numerical terms. In fact, mesophilic LABs can also grow on plate count agar at 35°C for 48 hours.

Line 94: ”.The producer of the Plate Count Agar, the city, and the country should be written in brackets.

Line 94: “Plate Count Agar at 35 °C for 48 h. S. aureus…..”. A full stop, after “48h” is necessary, because only S.aureus was counted according the ISO 6888-1:2021

Line 94: “S. aureus”. Please, write “Staphylococcus (S.) aureus

Line 98: please, write “Compendium of Methods for the Microbiological Examination of Foods”

Line 109: M17 agar is usually incubated at 37°C for 48 h

Line 113: “(3% hydrogen peroxide) and Gram tests, as well as morphologic observation”. These tests may not be sufficient for phenotypic LAB identification. I addition, it would be appropriate to indicate how the “morphological observation” was carried out.

Line 114: “…glycerol at -80 °C”.  The phrase “until further testing” (or similar) should be written.

Line 115: “2.4. Phenotypic characterization”. The paragraph 2.4 describes the “Antimicrobial activity of LAB isolates against three foodborne pathogens”, and “additional phenotypic assays” on a subset of LAB strains. The title of paragraph 2.4 should be changed.

Line 134:”……. around each LAB colony”. A reference number should be provided at the end of this sentence.

Lines 134, 160, 177, 196, and 208:  A reference number should be provided at the end of each sentence.

Lines 229 and 234: “The function principal ( )…”.  “….the pheatmap ( ) function”. Missing content within brackets.

Line 239: please, write “species”

Supplementary material. “Table 1: Descriptive statistics of the phenotypic characteristics by lactic acid bacteria specie”.

Please, write “species”.

Lines 240: "Leuconostoc mesenteroides". See "Instructions for authors" and referring to "Acronyms/ Abbreviations/ Initialisms"

Line 241: "Lactilactobacillus sakei" should be written. In addition, see also my comment in line 240.

Line 242: Correia Santos et (2017) [27] obtained the following results: “All products analyzed harbored enterococci, between 104 CFU/g for Catalao, Chouriço-preto and Linguiça and below 102 CFU/g for Salsichao and Paio”. Subsequently, these authors stated that: “In Portugal, fermented meat-sausages can present high counts of Enterococcus spp., with reports ranging from 104 to 108 CFU/g (Barbosa and others, 2009)”. In addition, the same authors referred to the "abundance of bacterial species" as "73 isolates for E. faecalis, 60 for E. faecium, and 14 for E. durans", and not to the abundance of CFU. Therefore, in my opinion, the authors should edit the phrase: “…..with counts ranging from  104 to 108 CFU/g…..”

Line 244: “Enterococcus (E)species” should be written

I ask authors: have their samples been smoked? If so, a comparison with the results of Correia Santos et al (2017) would be appropriate. In fact, these authors have sampled “Traditional dry smoked fermented meat sausages”.

Line 246: “…complex flavor profile of the alheira [27,28]”. Fraqueza (2015) [28] did not speak about “……lipid and protein hydrolysis, contributing to the complex flavor profile of the alheira….”. Therefore, the authors must check the Fraqueza’s article, and delete reference [28] from their text.

Line 247: “…and were found throughout the fermentation process [27]…”. Correia Santos et (2017) [27] do not speak about Leuconostoc spp., but only Enterococcus spp.

Line 248: “…and spanning 21 days”. This sentence refers to Moracanin et al.,2013 [28]. These authors report the fermentation process of the LAB (I, II. and III) for up to 21 days in samples of Sremska sausage.

Authors should change sentences in lines 247 - 248. In addition, they should give the correct references.

Line 255: “Weissela” must be modified as “ Weissella”. In addition, Franciosa et al (2021) [30] reported “Weissella”, not W.uvarum.

Lines 255-257: the authors write “Lactiplantibacillus plantarum, Pediococcus pentosaceus and Weissela uvarum were identified as less frequent LAB species, constituting 6.5, 4.8, and 1.6% of the total population, respectively”. The percentage for Weissella was not given. This sentence should be amended.

Line 257: “………total population, respectively.”. The reference number [30] should appear at the end of this sentence.

Lines 272-273: I would like to ask the author how the sentence in these lines relates to what is said in [33] “Safety assessment of foods derived from genetically modified microorganisms”. Report of a Joint FAO/WHO Expert Consultation on Foods Derived from Biotechnology (2001)

Line 277: “…..do not lead to a uniform collection of strains [1,34–37]”. Reference [37] is not strictly relate to this sentence.

Line 281: Patarata et al. (2008) have not studied the microbial flora from different Alheira samples. Reference [32] could be removed.

Figure 2: this figure has not an adequate resolution; it would be improved

Lines 298- 413: please, write “alheira” in italics.

Line 298: Worsztynowicz et al. (2019)[39] have not examined “alheira” samples. Reference [39] could be removed.

Line 300: Liu et al.(2021) [38] speak about sensory attributes of dry fermented sausages, but they do not take in consideration alheira samples.

Lines 306-308: authors state that “…The reduction of aw and pH during processing of the alheira contributes for the inhibition of pathogens development [1,3]”. Regarding physicochemical analyses, Ferreira et al. (2006) [1] report that “…pH, salt content and humidity per se, do not assure the microbiological safety of this product” (alheira). Esteves et al.(2008) [2] report that “….a product with intrinsic pH and water activity (Aw) characteristics which, alone or together, ensured the product’s stability at room temperature”, and  then “The drying/smoking process….inhibited, at least partially, the development of pathogenic bacteria”. In addition, these authors consider that “High microbian counts in the finished product could be explained by the use of unhygienic raw materials, substandard handling or by generally unsatisfactory hygienic conditions”. Neither Ferreira nor Esteves stated that the reduction of aw and pH contribute to the inhibition of pathogen development.

Authors would review the articles [1,3] and find other relevant references.

Liu et al 2021 [38] consider that “Lower pH can inhibit the growth of harmful bacteria and contribute to product safety,

and extend the shelf life of the product”, referring to fermented sausages.

Authors should check their sentences as well as the content of the reference [1] (lines 306-308).

Lines 314-415: authors state that “strong correlation was noted between mesophiles and LAB (presumptive Lactobacillus and Lactococcus), consistent with common attributes in dry fermented sausages [38]….”

In Liu et al.(2021) [38] you can observe the evolution of microbial populations during the ripening of different samples of sausages. Only Figure 3 (graphs B, C and D) shows that the trend in Total Bacterial Count (TB) could be correlated with LAB. I don't think that "the strong correlation between mesophiles and LAB" is strictly consistent with the common characteristics found in dry fermented sausages.

Lines 316-317: “highlighting the impact of alterations in meat properties on parallel changes in the microbiological composition”. What is the meaning of “alterations in meat properties”? How do these “alterations” correlate with Alheira's microbial composition? No samples alteration was described in the manuscript. The consideration given in lines 316-318 is not clear enough.

Lines 337-339: Franz et al (2003) [37] report that “Enterocins A and B from E. faecium CTC492, when added as semi-pure preparations, showed a marked antilisterial activity in model meat and meat products such as cooked ham, minced pork meat, deboned chicken breasts, pâte´ and ‘espetec’”, and “In contrast, Callewaert et al. (2000) showed that two bacteriocin-producing strains of E. faecium effectively inhibited a strain of L. innocua in model Spanish-style dry fermented sausage”.Hugas (1998) [40]  write “Enterococcus faecium CTC492 isolated from Spanish slightly fermented sausages and Ent. faecium DCP1146 isolated from Irish dairy products (Parente and Hill, 1992a; O’Keefe et al., 1996) produce the same bacteriocin, named enterocin A and enterocin 1146 respectively. Enterocin A inhibits other Lactobacillus, L. monocytogenes, C. divergens, Ent. faecalis and C. perfringens. recently it has been shown that E. faecium T136 produces enterocin A and B, while E. faecium P13 produces enterocin P and E. faecium L50 produces two novel enterocins L50A and LSOB (Cintas et al., 1998) all of them with a wide antimicrobial spectrum”.

It is advisable to consider the two references more precisely.

Lines 430-341: “….lower compared to those of other  LAB,….”. A reference should be written at the end of this sentences.

Line 341: Worsztynowicz et al (2019) [39] write: “……the extracellular protease produced by E. faecalis subsp. liquefaciens was able to hydrolyze casein, α-lactoalbumin, β-lactoglobulin and bovine serum albumin”.

Lines 341-342: please, write “although previous research has shown that by the action of enzymes able to hydrolyze casein, bovine serum albumin and β-lactoglobulin [39],….”. In addition, this phrase should be checked because it seems to be incomplete.

Line 344: please, write: “……local cuisine and heritage of the region [37]”.

Line 373: reference [41] could be integrated by one more recent article.

Line 443: “….pathways, including proteolysis .” At the end of this sentence, a reference should be given.

Line 458: A standardized approach could include the use of selected starter LABs. This could help to reduce both batch-to-batch and quality variability.

The correlation between the abstract and the content's conclusions should be improved.

REFERENCES: they must be checked and modified according to the “Instructions for Authors”

[15] - [19]- [22] -[24]  are available in PDF format; the authors can write the reference according to the “Instructions for Authors”

[26]: Authors should check this reference and improve it, except for website citation.

Comments on the Quality of English Language

Minor editing of English language are required.

Round 2

Reviewer 2 Report

Comments and Suggestions for Authors

Although the revised version of the manuscript is anything but satisfactory to my concerns, but it seems that other reviewers’ comments may help the authors to improve the manuscript with minimum requirement.

However, there are strange repeats in the revised manuscript, e.g., line369–374, 512–521, the authors have to correct them properly.

Author Response

we have double-checked the revised manuscript document and could not find the errors mentioned. We believe this could be attributed to an error at the compilation level in the mdpi platform.

Reviewer 3 Report

Comments and Suggestions for Authors

The authors have successfully addressed this reviewer’s primary concern which was whether sufficient information was provided to allow reproduction of the experimental trial.  In addition, many of the editorial issues appear to have been corrected.  However, in the copy of the revised manuscript that was downloaded, there were periodic repeats of lines of text.  I have not attempted to articulate each these occurrences, assuming that the journal’s copy editors will deal with these errors. 

Comments on the Quality of English Language

Much improved.  There was a series of repeating lines in the manuscript that appear to be a problem associated with the upload.

Author Response

We thank the reviewer for this comment, we have double-checked the revised manuscript document and could not find the errors mentioned. We believe this could be attributed to an error at the compilation level in the mdpi platform.